# Exposure to *Bacillus cereus* in Water Buffalo Mozzarella Cheese

**DOI:** 10.3390/foods9121899

**Published:** 2020-12-19

**Authors:** Angela Michela Immacolata Montone, Federico Capuano, Andrea Mancusi, Orlandina Di Maro, Maria Francesca Peruzy, Yolande Thérèse Rose Proroga, Daniela Cristiano

**Affiliations:** 1Department of Food Microbiology, Istituto Zooprofilattico Sperimentale del Mezzogiorno, Via Salute 2, 80055 Portici (NA), Italy; angela.montone@izsmportici.it (A.M.I.M.); federico.capuano@cert.izsmportici.it (F.C.); andrea.mancusi@izsmportici.it (A.M.); orlandina.dimaro@izsmportici.it (O.D.M.); daniela.cristiano@izsmportici.it (D.C.); 2Department of Veterinary Medicine and Animal Production, University of Naples “Federico II”, Via Delpino 1, 80137 Naples, Italy; mariafrancesca.peruzy@unina.it

**Keywords:** *Bacillus cereus*, water buffalo mozzarella cheese, risk of poisoning, toxinotype

## Abstract

*Bacillus cereus* is a spoilage bacterium and is recognized as an agent of food poisoning. Two food-borne illnesses are caused by *B. cereus*: a diarrheal disease, associated with cytotoxin K, hemolysin BL, non-hemolytic enterotoxin and enterotoxin FM, and an emetic syndrome, associated with the cereulide toxin. Owing to the heat resistance of *B. cereus* and its ability to grow in milk, this organism should be considered potentially hazardous in dairy products. The present study assessed the risk of *B. cereus* poisoning due to the consumption of water buffalo mozzarella cheese. A total of 340 samples were analyzed to determine *B. cereus* counts (ISO 7932:2005); isolates underwent molecular characterization to detect the presence of genes encoding toxins. Eighty-nine (26.1%) samples harbored *B. cereus* strains, with values ranging from 2.2 × 10^2^ to 2.6 × 10^6^ CFU/g. Isolates showed eight different molecular profiles, and some displayed virulence characteristics. Bacterial counts and the toxin profiles of isolates were evaluated both separately and jointly to assess the risk of enteritis due to *B. cereus* following the consumption of buffalo mozzarella cheese. In conclusion, the results of the present study showed that the risk of poisoning by *B. cereus* following the consumption of this cheese was moderate.

## 1. Introduction

The *Bacillus cereus* group, also known as *B. cereus sensu lato* (*s.l.*), comprises several closely related species of Gram-positive aerobic spore-forming bacteria, which are widely distributed in nature. Within the group, *B. cereus sensu stricto* (*s.s.*), *B. cereus*, *B. thuringiensis*, *B. anthracis*, *B. mycoides*, *B. pseudomycoides*, *B. weihenstephanensis*, *B. cytotoxicus* and *B. toyonensis* are the most important members [1,2,3,4]. The majority of these species are usually harmless. However, some species, especially *B. cereus s.s.*, can cause food-borne illness in humans. *B. cereus s.s.*, according to the combination of genes harbored, and therefore to the toxins produced, can cause two different types of food poisoning: a diarrheal syndrome and an emetic syndrome [5].

The *B. cereus* diarrheal syndrome is caused by several heat-labile and gastro-labile toxins, including hemolysin BL (HBL), non-hemolytic enterotoxin (NHE), cytotoxin K (CytK) and enterotoxin FM (EntFM) [6,7,8,9,10], which are produced during vegetative bacterial growth in the small intestine. By contrast, the emetic syndrome is caused by the production of the heat-stable and gastro-stable peptide toxin cereulide (Ces) [11], which is produced in foodstuffs.

Several other putative enterotoxins (hemolysin II, hemolysin III, cereolysin AB, cereolysin O, and enterotoxin T) that might also contribute to virulence have been described [12]. Moreover, it has been reported that genes encoding toxins may characterize species other than *B. cereus sensu stricto* [13,14]. Therefore, other species comprising the *B. cereus* group may also be pathogenic for human beings. Detection of the genes, and also of the toxins, provides an indication of the toxicity of a strain, but is not sufficient to fully distinguish the virulence profiles of different *B. cereus* strains [12]. Thus, to assess virulence correctly, other factors (e.g., the infective dose) should be included in the analysis [11].

In the literature, a value of 10^5^–10^8^ cells or spores per gram is generally considered to be the infective dose for *B. cereus* [15,16] but lower bacterial counts have also been reported in cases of human poisoning; therefore, no food containing more than 10^3^ CFU/g can be considered completely safe for consumption [17].

*Bacillus* spp.related outbreaks in Europe have increased significantly. However, in Italy, notifications and reports of *B. cereus* poisoning are scarce [18,19,20]. Therefore, the true impact of *B. cereus* poisoning on health is still not known.

Although most human cases have been associated with the consumption of rice and vegetables, almost all kinds of food may be involved in food-borne *B. cereus* poisoning [17,21,22]. Moreover, *B. cereus* is a frequent contaminant of raw milk and, owing to the extreme resistance of endospores to milk pasteurization, cheese-processing heat treatments and fermentation, it has also been frequently reported in dairy products [8,23].

Water buffalo mozzarella cheese (WBMC) is one of southern Italy’s most popular “pasta filata” cheeses, and is globally commercialized, usually at refrigeration temperature [24]. During the manufacture of WBMC, the curd is stretched in hot water (about 90 °C), which reduces the microbial load; nevertheless, it has been demonstrated that *Salmonella typhimurium* and *Listeria monocytogenes* can survive [25,26]. This suggests that the survival of spore-forming bacteria is even more likely. Although studies on the contamination of cheeses by *B. cereus* have been conducted [27,28,29], little is known about its presence in WBMC. Therefore, the aims of this study were: (i) to study the occurrence of *B. cereus s.l.* in WBMC; (ii) to evaluate the presence of nine genes encoding toxins in the isolated strains; and (iii) to assess the risk of *B. cereus* poisoning due to the consumption of water buffalo mozzarella cheese through a combined approach based on the evaluation of bacterial contamination and analysis of the virulence profile.

## 2. Materials and Methods

### 2.1. Sampling

From January 2015 to October 2017, a total of 340 WBMC samples, with a shelf life no longer than seven days, were aseptically collected at different dairies in the Campania region of southern Italy. Samples were transported at 4 °C to the laboratory and immediately analyzed.

### 2.2. Bacterial Counting and Identification

Microbiological assays were performed in accordance with the ISO 7932:2005 method. In brief, 10 g of each sample and 90 mL (1:10 (*w*/*w*)) of sterilized buffered peptone water (BPW; Oxoid, UK) were placed in a sterile stomacher bag and homogenized for 3 min at 230 rpm using a peristaltic homogenizer (BagMixer^®^400 P, Interscience, Saint Nom la Bretèche, France). Subsequently, the homogenized samples were submitted to ten-fold serial dilutions in BPW followed by spread-plating on Mannitol Egg Yolk Polymyxin Agar (MYP; Thermo Scientific, Waltham, MA, USA) and incubated at 30 °C for 48 h. Suspected colonies were counted and randomly picked out (up to five isolates) for the evaluation of hemolytic activity on Trypticase Soy Agar 5% sheep blood (Biolife, Monza, Italy). Confirmatory biochemical assessment of the isolates was carried out by means of a Vitek device (bioMerieux, France), according to the manufacturer’s instructions. Harvested colonies were then subcultured in nutrient broth (Biolife, Monza, Italy), and incubated at 30 °C for 4 h for subsequent real-time PCR.

### 2.3. Estimation of the Probability of Growth

To estimate *B. cereus* growth rates at three different temperatures (15, 18 and 22 °C), the ComBase Predictor software (www.combase.cc) was used. The parameters considered (pH = 5.7 and a_w_ = 0.997) were those described by [30]. The starting counts used in the predictions were 2 and 3 log CFU/g, while for the physiological state, the input parameter (0.00027) contained in the ComBase Predictor was chosen.

### 2.4. Gene Detection in B. cereus Strains

One milliliter of each broth culture was transferred into a clean microcentrifuge tube and centrifuged for 3 min at 4000× *g*. Subsequently, the supernatant was discarded, and the pellet was washed twice with 500 μL of sterile DNase-free water. After a second centrifugation, the supernatant was discarded, and the pellet was resuspended in 100 μL of Milli-Q. The suspension was then incubated at 98 ± 1 °C for 10 min and centrifuged at 10^4^× *g* for 5 min.; 2.5 µL of the DNA extracted was used as a template for producing amplicons in a conventional PCR reaction, which was performed in a final volume of 25 µL using the GoTaq Green Master Mix (Promega, Madison, WI, USA). Nine genes were investigated: *hblA*, *hblD* and *hblC*, encoding HDL enterotoxin; *nheA*, *nheB* and *nheC*, encoding NHE; *cyfK*, encoding CytK; *entFM*, encoding EntFM; and *ces*, encoding Ces. The primers used in the present study are listed in Table 1.

The PCR products were highlighted by means of a QIAxcel Advanced device (Qiagen). *Bacillus cereus* ATCC 11778 PK/5, *Bacillus cereus* ATCC 14579, and *B. thuringiensis* subsp. *aizawaii* strain GC-91, were used as controls. For confirmatory purposes, the amplicons obtained for cereulide detection were sequenced and compared with the sequences in Gene Bank.

### 2.5. Exposure Assessment

To estimate the probability that buffalo mozzarella consumers will suffer an adverse effect because of *B. cereus* contamination, the exposure assessment was performed in the final phase of the food chain, i.e., at retail outlets. The bacterial count and the virulence profile (toxinotype) of the isolates were evaluated both separately and jointly. Taking into account the estimated average annual consumption of mozzarella and the percentage of samples contaminated (prevalence), the exposure risk was calculated as:R_xp_ = (M_C_/100) × R_c_(1)
where: 

R_xp_ = exposure risk;M_C =_ mozzarella per capita average consumption; R_c_ = risk classes (prevalence).

## 3. Results

### 3.1. Bacterial Counts

*B. cereus* was recovered in 89 of the 340 samples (26.2%), with a bacterial count ranging from 2.2 × 10^2^ to 3.6 × 10^6^ CFU/g (Table 2). Positive samples were stratified into five classes on the basis of their *B. cereus* count (Table 2). Overall, 59 of the 89 positive strains (66.3%) showed a level of contamination <10^3^ CFU/g. The other positive strains showed higher levels of contamination (10^3^–10^4^ (20.2%); 10^4^–10^5^ (6.7%); 10^5^–10^6^ (3.4%); >10^6^ (3.4%)).

Specifically, 6 (1.8%), 24 (7.1%) and 59 (17.3%) samples showed levels of contamination >10^5^, between 10^5^ and 10^3^, and <10^3^ CFU/g, respectively.

### 3.2. Estimation of the Probability of Growth

*B. cereus* in samples with initial contamination levels of 2 log CFU/g and 3 log CFU/g, under the conditions considered (storage temperature (15 °C; 18 °C; 22 °C), pH (5.7) and aw (0.997)), reached levels of ≥ 5 log CFU/g in 21 to 53 h (Table 3). Specifically, in samples with an initial contamination level of 2 log CFU/g, *B. cereus* reached levels of ≥5 log CFU/g in around 25, 41 and 63 h at 22, 18 and 15 °C, respectively. In those with an initial contamination level of 3 log CFU/g, *B. cereus* reached levels of ≥5 log CFU/g in around 21, 35 and 53 h at 22, 18 and 15 °C, respectively. Moreover, regarding microbial growth, the maximum growth rate (log·conc/h), the doubling time (hours), the maximum population density (MPD; log CFU/g) and the lag time (hours) were estimated (Appendix A). On comparing the three temperatures, at 22 °C *B. cereus* showed the highest growth rate per hour (0.268 log·conc/h), the shortest doubling time (1.125 h) and the shortest lag time (13.29 h).

### 3.3. Gene Detection in B. cereus Strains

A total of 89 strains were analyzed for gene detection. The most common genes in *B. cereus* strains were *entFM* and *nheC* (97.7% each), followed by *nheA* (95.5%), *nheB* (89.9%), *cytK* (46.1%), *hblA*, *hblC*, *hblD* (37.1% each) and *ces* (1.1%) (Table 4).

The three genes (*hblA*, *hblC*, *hblD*) coding for the trimeric HBL toxin were always simultaneously detected. The concomitant presence of the three genes coding for the NHE toxin was observed in 80 isolates; five isolates were positive for both the *nheA* and *nheC* genes, and two were positive exclusively for the *nheC* gene. Only one isolate harbored the *Cereulide Synthetase* gene. Moreover, eight haplotypes (Table 2) were identified. Overall, 24.72% of the strains harbored *nhe* (*A*, *B*, *C*), *hbl* (*A*, *C*, *D*), *cytK* and *entFM* genes (haplotype H1). The haplotypes H2 (*nhe* (*A*, *B*, *C*); *hbl* (*A*, *C*, *D*)) and H3 (*nhe* (*A*, *B*, *C*); *cytK*; *entFM*) were identified in 6.74% and 21.35% of isolates, respectively, while the haplotypes H4 (*nhe* (*A*, *B*, *C*); *entFM*), H5 (*nhe* (*A*, *B*, *C*); *entFM*; *ces*) and H6 (*nhe* (*A*, *C*); *hbl* (*A*, *C*, *D*); *entFM*) were identified in 35.95%, 1.12% and 5.62% of isolates, respectively.

### 3.4. Exposure Assessment

The average per capita consumption of cheese in Italy is 23 kg (ranging from 21.5 kg to 25 kg) [33,34], with mozzarella accounting for about 15% of the total. The risk of exposure (R_xp_ = (M_C_/100) × R_c_) was calculated on the basis of both the estimated average annual consumption of mozzarella (M_C_ = 3.5 kg per capita) and of the prevalence of contamination observed in the present study (R_c_ = Table 5). The estimated exposure risk varied according to the analytical method adopted (pathogen contamination, virulence profile and a combined approach based on both bacterial contamination and the virulence profile). If only bacterial contamination is considered, the exposure risk is “high”, “moderate” and “low” for people whose annual consumption is around 50, 250 and >600 g of mozzarella, respectively. (Figure 1). If only the molecular profile (toxinotyping) is considered, the exposure risk is “high”, “moderate” and “low” for people whose annual consumption is around 200, 250 and 400 g, respectively. On combining bacterial count and the virulence profile, the exposure risk is “high” and “moderate” for people whose annual consumption is around 100 and 200 g, respectively.

## 4. Discussions

In the present study, 89 samples were found to be contaminated by *B. cereus*, 6.74% of which showed levels of contamination considered to constitute a significant risk of poisoning (>5 Log CFU/g) [16] (Table 2) already at the moment of purchase. To our knowledge, this is the first study to provide data on the prevalence of *B. cereus* in WBMC. The presence of this pathogen in these samples may be due either to the germination of spores present in raw milk which were not suppressed during the phase of curd stretching, or to the survival of the pathogen due to protection by the protein matrix and fat content and/or to an uneven distribution of heat during the curd-stretching process [25].

Regarding the risk of poisoning, storage conditions at home may also have a heavy impact. Indeed, our estimates of the growth rate of *B. cereus* in WBMC samples with an initial contamination of 2 or 3 log CFU/g under the conditions considered (15, 18, 22 °C) revealed that the pathogen can reach risk levels (≥5 log CFU/g) in 21 to 63 h (Table 3). Storage conditions, in terms of temperature and time, as commonly indicated by producers, may differ considerably; however, in the geographical area of production, WBMC is usually stored at room temperature and eaten within a few days after its production. To simulate these storage conditions, we considered three temperatures (15, 18 and 22 °C) in our analysis. The short shelf life of mozzarella contributes to reducing the risk, although appropriate storage temperatures must be ensured, especially in summer. Indeed, at the storage temperature of 22 °C, contaminated samples can reach a risk condition (>5 log CFU/g) in approximately 24 h (Table 3). The increase in bacterial load during storage spoils the cheese, thereby reducing the risk to health. By contrast, in dairy products that show a high level of contamination from the outset, significant alterations do not occur; in these products, the risk is potentially greater.

The EU Commission’s definition of exposure assessment is “… estimation of how likely it is that an individual or a population will be exposed to a microbial hazard and what numbers of micro-organisms are likely to be ingested” [35]. However, the assumption that the risk depends basically on the level of microbial contamination does not consider the various molecular profiles of *B. cereus* isolates. Indeed, *B. cereus* is potentially able to synthesize several toxins, but most of the bacteria belonging to this group lack toxin-producing genes. To assess risk correctly, it is therefore crucial to evaluate the molecular profile (toxinotype) of the isolates, rather than simply counting the bacteria [11]. In the present study, the presence of nine genes encoding toxins was evaluated in the strains isolated. The analysis of the isolates showed that not all of them were positive for all the genes tested for. However, most of the isolates analyzed in our study harbored one or more genes involved in the production of toxins (Table 2).

In the present study, the isolates were classified into hazard categories according to their molecular profile (Table 5). On the basis of the genes harbored, six haplotypes (H1–H6) showed virulence properties related to possible food-borne poisoning, as reported by [36]. Based on the presence or absence of the different genes recognized as playing a major role in the disease, we have categorized the haplotypes in three groups on the basis of the risk for *B. cereus* disease: haplotypes at “high risk”, “moderate risk” and “low risk”. The haplotype H1 harboring *nhe* (*A*, *B*, *C*), *hbl* (*A*, *C*, *D*), *cytK* and *entFM* genes has been considered the haplotype as having the greatest risk (“High Risk”) to cause enteric disorders.

Isolates characterized by the haplotypes H2 (*nhe* (*A*, *B*, *C*); *hbl* (*A*, *C*, *D*)) and H3 (*nhe* (*A*, *B*, *C*); *cytK*; *entFM*) showed a “Moderate Risk” owing to the presence of the *nhe* and *hbl* genes in H2 and the presence of *nhe* and *cytK* genes in H3. However, compared with H1, the risk of enteric disorders arising from the consumption of foods contaminated with H2 and H3 is lower for the absence of *cytK* gene, which toxin is highly toxic towards human intestinal epithelial cells [37], or *hbl* genes. Indeed, HBL complex is one of the most important responsible for enterotoxigenic activity of *B. cereus* strains, thus its absence leads to a reduction in the cytotoxic and hemolytic activity of this pathogen [38,39]. Haplotypes H4 (*nhe* (*A*, *B*, *C*); *entFM*), H5 (*nhe* (*A*, *B*, *C*); *entFM*; *ces*) and H6 (*nhe* (*A*, *C*); *hbl* (*A*, *C*, *D*); *entFM*) showed a lower level of risk (“Low Risk”), owing to the absence in all of them of the *cytK* gene and the additional absence of genes coding for HBL toxin components or of the *nhe*B gene. H6, which was infrequently detected (5.6%), harbored the incomplete *nhe* operon; it was therefore regarded as posing a low risk, because all three components (A, B and C) of the *nhe* operon are needed for full toxicity [11]. The *ces* gene was detected in only one *B. cereus* isolate, jointly with the complete *nhe* operon and *entFM* gene (haplotype 5). As reported in Table 5, considering the virulence profile alone, 22 (6.5%), 25 (7.3%) and 38 (11.2%) of the *B. cereus* isolates analyzed showed a profile of high, moderate, and low risk, respectively.

However, risk cannot be reliably assessed by considering the molecular profile and the bacterial count separately [11]. In the present study, risk was assessed by jointly evaluating the results of microbiological and molecular assays. In this analysis, the “high risk” category comprised samples with *B. cereus* contamination above 10^3^ CFU/g—an infective dose considered unsafe for human consumption [40]—and high-risk toxinotype strains (haplotype 1), and those with higher *B. cereus* contamination (>10^5^ CFU/g) but lower molecular risk profiles of the isolates (haplotypes 2 and 3). The “moderate risk” group comprised samples displaying a higher risk of toxic strains (haplotype 1) but lower bacterial contamination, and samples with an intermediate level of contamination (from 3 to 5 log CFU/g) and haplotypes 2 and 3. All the remaining samples were classified as “no risk”, as shown in Table 5.

According to the above risk classification, most of the samples showed no risk (90.0%). By contrast, bacterial counting and the molecular approach assigned fewer samples to this group (73.8% and 75%, respectively).

However, in the present study, the occurrence of the toxins in WBMC was not evaluated. Unlike the diarrheal form caused by the enterotoxins that are produced in the intestine, the emetic syndrome is associated with the presence of the toxin cereulide pre-formed in foods [11]. Thus, for a complete virulence assessment, the detection of the toxin cereulide should be also included.

The response of consumers to exposure to *B. cereus* is highly variable and depends on many factors, including the virulence profile of the pathogen, the numbers of cells ingested as spores or as vegetative cells [41], the general health status of the host, and the host’s gut microbiome. If the risk of exposure were estimated only on the basis of the intensity of exposure, all these aspects would have little impact on the assessment. Moreover, to appraise risk, data on mozzarella consumption by different subpopulations should be included in the analysis; unfortunately, however, little information on the typology of mozzarella consumers is available.

On the basis of the results of the present study, given that a standard portion of mozzarella is of about 100 g, the annual risk of exposure is assessed as being from 0.5 to 2.9 portions at high risk, from 2.5 to 3.9 portions at moderate risk and from 0.5 to 6 portions at low risk. Overall, analysis of the toxin profile alone leads to an overestimation of the risk, while evaluations based on the *B. cereus* count and the combined assessments of the count and the toxinotype provide more homogeneous results; this latter approach can explain the reported cases of poisoning associated with the consumption of foods with low levels of *B. cereus* contamination [17]. According to this approach, in order to engender a high risk of poisoning, the total number of *B. cereus* cells should be between 10^5^/CFU × 100 g and >10^7^/CFU × 100 g, depending on the characteristics of the microorganisms.

## 5. Conclusions

In conclusion, the consumer runs a moderate risk of buying water buffalo mozzarella cheese with a level of *B. cereus* high enough to cause food poisoning. However, the results of the present study showed that the risk of poisoning may increase during storage at room temperature, because the pathogen is able to grow rapidly. Moreover, a high percentage of the strains isolated harbored genes involved in toxin production, and a high percentage showed virulence properties associated with possible food-borne poisoning. However, on combining the findings of bacterial counting and analysis of the molecular profile, a different level of risk emerged. Specifically, a lower number of samples proved to be at high or moderate toxic risk. Furthermore, if the average per capita consumption of water buffalo mozzarella cheese is considered, the risk of poisoning by *B. cereus* is even lower.

Only a combined approach based on the *B. cereus* count and the molecular profiles of isolates can accurately assess the risk of poisoning. However, for a more complete evaluation, data on the typology of mozzarella consumers and the impact of *B. cereus* poisoning on human beings would be necessary. Indeed, the present study lacks quantitative information on the consequences of exposure. This is because the surveillance of enteric disorders in human beings is in some cases inadequate, especially with regard to self-limiting disorders, such as those caused by *B. cereus*. Moreover, additional research is clearly needed in order to investigate the occurrence of the toxins in the water buffalo mozzarella cheese for a deep evaluation of the health risk originating from the consumption of this food.

## Figures and Tables

**Figure 1 foods-09-01899-f001:**
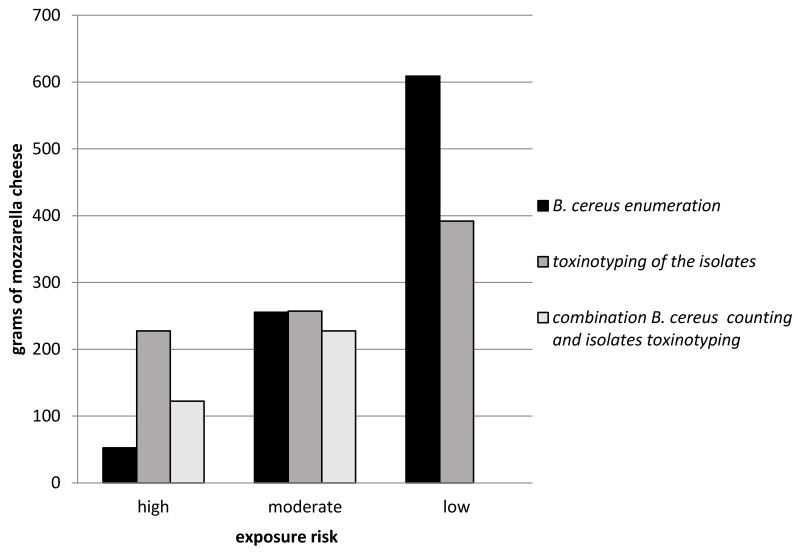
Annual amount of mozzarella consumed, subdivided by level of risk (high; moderate; low) and the method of risk determination (*B. cereus* counting; molecular profiling of isolates; combined evaluation: *B. cereus* counting plus molecular profiling of isolates).

**Table 1 foods-09-01899-t001:** Primer sequences utilized to detect nine genes encoding five toxins in *B. cereus* isolates.

Toxin	Target Gene	Primer Sequence (5′–3′)	Product Size (bp)	Reference
HBL	*hbl*A	F-GCAAAATCTATGAATGCCTA	884	[31]
		R-GCATCTTGTTCGTAATGTTTT		
	*hbl*D	F-GAAACAGGGTCTCATATTCT	1018	
		R-CTGCATCTTTATGAATATCA		
	*hbl*C	F-CCTATCAATACTCTCGCAA	695	
		R-TTTCCTTTGTTATACGCTGC		
NHE	*nhe*A	F-TAAGGAGGGGCAAACAGAAG	759	
		R-TGAATGCGAAGAGCTGCTTC		
	*nhe*B	F-CAAGCTCCAGTTCATGCGG	935	
		R-GATCCCATTGTGTACCATTG		
	*nhe*C	F-ACATCCTTTTGCAGCAGAAC	618	
		R-CCACCAGCAATGACCATATC		
CytK	*cyt*K	F-CGACGTCACAAGTTGTAACA	565	
		R-CGTGTGTAAATACCCCAGTT		
EntFM	*entFM*	F-GTTCGTTCAGGTGCTGGTTAC	486	
		R-AGCTGGGCCTGTACGTACTT		
Ces	*ces*	F-GGTGACACATTATCATATAAGGTG	1271	[32]
		R- GTAAGCGAACCTGTCTGTAACAACA		

**Table 2 foods-09-01899-t002:** *B. cereus* counts in 340 water buffalo mozzarella cheese (WBMC) samples and haplotypes of isolated strains. Positive samples were stratified into five classes based on the *B. cereus* count.

				*B. cereus* CFU/g				
Haplotypes	Molecular Profile	Negative Samples	Positive Samples	<10^3^	10^3^–10^4^	10^4^–10^5^	10^5^–10^6^	>10^6^
H1	*nhe(A,B,C); hbl (A,C,D); cytK; entFM*	-	22	13	5	1	2	1
H2	*nhe(A,B,C); hbl (A,C,D)*	-	6	3	2	1	-	-
H3	*nhe(A,B,C); cytK; entFM*	-	19	10	5	1	1	2
H4	*nhe(A,B,C); entFM*	-	32	27	3	2	-	-
H5	*nhe(A,B,C); entFM; ces*	-	1	-	-	1	-	-
H6	*nhe (A,C); hbl (A,C,D); entFM*	-	5	3	2	-	-	-
H7	*No genes detected*	-	2	2	-	-	-	-
H8	*nheC; entFM*	-	2	1	1	-	-	-
		251	89	59	18	6	3	3

**Table 3 foods-09-01899-t003:** Rate of growth of *B. cereus* from initial contamination levels of 2 log CFU/g and 3 log CFU/g in water buffalo mozzarella cheese (pH 5.7 and a_w_ 0.997) at three temperatures (15, 18 and 22 °C) calculated by ComBase Predictor software (www.combase.cc).

	Concentration (Log Cells/g) at Different Temperatures					
Time (Hours)	22 °C		18 °C		15 °C	
	Log CFU/g	Log CFU/g	Log CFU/g	Log CFU/g	Log CFU/g	Log CFU/g
0.00	3.00	2.00	3.00	2.00	3.00	2.00
20.62	5.01	4.01	3.2	2.2	3.01	2.02
24.60	6.06	5.07	3.55	2.55	3.04	2.04
34.73	7.58	7.4	5.03	4.04	3.35	2.35
40.88	7.6	7.6	6.01	5.02	3.81	2.81
53.19	7.6	7.6	7.46	6.9	5.02	4.03
62.59	7.6	7.6	7.6	7.56	6.00	5.01

**Table 4 foods-09-01899-t004:** Number (No.) and percentage (%) of *B-cereus* strains containing nine genes (*hblA*, *hblC*, *hblD*, *nheA*, *nheB*, *nheC*, *cytK*, *entFM* and *ces*) encoding five toxins (HBL, NHE, CytK, EntFM and Ces).

	Gene Target								
	*hbl*			*nhe*			*cytK*	*entFM*	*ces*
	*A*	*C*	*D*	*A*	*B*	*C*			
No.	33	33	33	85	80	87	41	87	1
%	37.1	37.1	37.1	95.5	89.9	97.7	46.1	97.7	1.1

**Table 5 foods-09-01899-t005:** Number and percentage of water buffalo mozzarella cheese samples falling into the four risk categories (High, Moderate, Low, and No Risk) according to the methods of determination adopted (pathogen contamination, virulence profile and a combined approach based on both bacterial contamination and virulence profile).

	Degree of Risk			
	High	Moderate	Low	No Risk
*B. cereus* contamination	>5 log CFU/g	>3 log to 5 log CFU/g	<3 log CFU/g	No *B. cereus* isolate
No. mozzarella samples	6 (1.8%)	24 (7.1%)	59 (17.3%)	251 (73.8%)
Virulence profile of isolates	*haplotype 1*	*haplotypes 2* and *3*	*haplotypes 4*, *5* and *6*	Negative samples, and isolates without any virulence determinants
No. mozzarella samples	22 (6.5%)	25 (7.3%)	38 (11.2%)	255 (75%)
Combination: bacterial contamination and isolate virulotype	>3 log CFU/g and *haplotype 1* >5 log CFU/g and *haplotypes 2* and *3*	<3 log CFU/g and *haplotype 1* from 3 log to 5 log CFU/g and *haplotypes 2* and *3*	>5 log and *haplotypes 4*,*5* and *6*	All remaining conditions
No. mozzarella samples	12 (3.5%)	22 (6.5%)	0 (0.0%)	306 (90.0%)

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
