# Peer review of "Exposure to *Bacillus cereus* in Water Buffalo Mozzarella Cheese"

_foods, 2020, doi:10.3390/foods9121899_

Round 1
Reviewer 1 Report
The study describes the occurrence of B. cereus in mozzarella cheese, estimates some virulence markers and tries to calculate an exposure risk. In principle the data generated are new and interesting, however, there are a number of serious drawbacks:
In general, the manuscript looks rather like a draft and not like a manuscript ready for publication. It needs thorough revision of English language, parts of the paper (discussion and conclusions) are hardly understandable. Bacterial names should be in italics.
Specific comments:
Introduction (and discussion): Literature cited is not up to date, key findings about virulence published after 2015 are missing.
Materials and Methods: Due to too many inaccuracies and mistakes, it is not possible to guess if the experiments were conducted properly. The experiments must be explained in much more detail, particularly 2.3 to 2.5. Are there references for the primers (Table 1)?
Results: Description of results is quite superficial, it consists mainly of tables.
Table 2 (and text): Names of genes should be uncapitalized.
Table 3: It is not necessary to give three decimal places for the time, why are such strange and variable intervals chosen from the calculation?
Table 4: Names of genes should be uncapitalized.
Paragraph 3.4 and Fig. 1 are not understandable, should be explained in much more detail.
The results presented do not support the conclusions drawn, particularly regarding the “dangerousness” of the haplotypes, and especially as current literature is ignored.
Discussion: Beside the general comment, the main issue is the definition of the “hazard categories”:
The reference cited does not provide information on this topic. The definition seems quite arbitrary, why are strains harboring nhe genes and entFM (H4) more dangerous than strains with hbl and entFM (H6)? Strains able to produce complete NHE and/or HBL should be considered virulent, further differentiation needs testing for high and low toxin production.
Table 5: according to the above comment the “risk degrees” are not comprehensible, the data presented are not a risk assessment in the strict sense.
Conclusions are not supported by the results.
Author Response
Dear Reviewer,
We thank for the valuable comments.
We made the necessary adaptations and corrections throughout the manuscript (highlighted in yellow).
A detailed answer on each remark is included in the rebuttal. Moreover, based on the reviewer suggestion we also have the paper revised by a native English speaker (highlighted in green).
Yours sincerely,
Yolande Therese Rose Proroga, corresponding author
Response to Reviewer 1 Comments
The study describes the occurrence of B. cereus in mozzarella cheese, estimates some virulence markers and tries to calculate an exposure risk. In principle the data generated are new and interesting, however, there are a number of serious drawbacks:
POINT 1: In general, the manuscript looks rather like a draft and not like a manuscript ready for publication. It needs thorough revision of English language, parts of the paper (discussion and conclusions) are hardly understandable. Bacterial names should be in italics.
Response 1: The manuscript has now been improved basing on reviewer suggestion and moreover, now we also have the paper revised by a native English speaker.
POINT 2: Introduction (and discussion): Literature cited is not up to date, key findings about virulence published after 2015 are missing
Response 2: Thank you for this suggestion. The literature has been up to date.
POINT 3: Materials and Methods: Due to too many inaccuracies and mistakes, it is not possible to guess if the experiments were conducted properly. The experiments must be explained in much more detail, particularly 2.3 to 2.5. Are there references for the primers (Table 1)?
Response 3: The Material and Methods section has now been improved basing on the reviewer suggestion and the reference for the primers has been included in Table1.
POINT 4: Results: Description of results is quite superficial, it consists mainly of tables.
Response 4: Basing on the reviewer suggestion, the results section it has been extended by describing the results also in full in the text and not only by reporting them in the table.
POINT 5: Table 2 (and text): Names of genes should be uncapitalized.
Response 5: This was indeed a mistake. The name of genes is now in lowercase letters.
POINT 6: Table 3: It is not necessary to give three decimal places for the time, why are such strange and variable intervals chosen from the calculation?
Response 6: The decimal values have now been deleted.
POINT 7: Table 4: Names of genes should be uncapitalized.
Response 7: This was indeed a mistake. The name of genes is now in lowercase letters.
POINT 8: Paragraph 3.4 and Fig. 1 are not understandable, should be explained in much more detail.
Response 8: We agree with the remark and indeed the “Exposure assessment” was not well explained in the paragraph 3.4. The paragraph 3.4 has now been improved. Please refer to lines 186-198.
POINT 9: The results presented do not support the conclusions drawn, particularly regarding the “dangerousness” of the haplotypes, and especially as current literature is ignored.
Response 9: The current literature has now been included and the conclusion has been rewritten as:
“In conclusion, the consumer runs a moderate risk of buying water buffalo mozzarella cheese with a level of B. cereus high enough to cause food poisoning. However, the results of the present study showed that the risk of poisoning may increase during storage at room temperature, since the pathogen is able to grow rapidly. Moreover, a high percentage of the strains isolated harbored genes involved in toxin production, and a high percentage showed virulence properties associated with possible food-borne poisoning. However, on combining the findings of bacterial counting and analysis of the molecular profile, a different level of risk emerged. Specifically, a lower number of samples proved to be at high or moderate toxic risk. Furthermore, if the average per capita consumption of water buffalo mozzarella cheese is considered, the risk of poisoning by B. cereus is even lower.
Only a combined approach based on the B. cereus count and the molecular profiles of isolates can accurately assess the risk of poisoning. However, for a more complete evaluation, data on the typology of mozzarella consumers and the impact of B. cereus poisoning on human beings would be necessary. Indeed, the present study lacks quantitative information on the consequences of exposure. This is because the surveillance of enteric disorders in human beings is in some cases inadequate, especially with regard to self-limiting disorders, such as those caused by B. cereus. “ Please refer to lines 304-322.
POINT 10: Discussion: Beside the general comment, the main issue is the definition of the “hazard categories”
Response 10: As suggested by the reviewer the discussion has been improved. Regarding the “ hazard categories” the discussion have been modified as
“In the present study, the isolates were classified into hazard categories according to their molecular profile (Table 5). On the basis of the genes harbored, six haplotypes (H1-H6) showed virulence properties related to possible food-borne poisoning, as reported by [36]. One haplotype harbored nhe (A, B, C), hbl (A, C, D), cytk and entfm genes and exhibited the greatest risk (“High Risk”) of enteric disorders.
Isolates characterized by the haplotypes H2 [nhe (A,B,C); hbl (A,C,D)] and H3 [nhe (A,B,C); cytk; entfm] showed a “Moderate Risk”. Haplotypes H4 [nhe (A,B,C); entfm], H5 [nhe (A,B,C); entfm; ces] and H6 [nhe (A,C); hbl (A,C,D); entfm] showed a lower level of risk (“Low Risk”), owing to the absence either of genes coding for Hbl toxin components or of the nheB gene. H6, which was infrequently detected (5.6%), harbored the incomplete Nhe operon; it was therefore regarded as as posing a low risk, since all three components (A, B and C) of the Nhe operon are needed for full toxicity [11]. The ces gene was detected in only one B. cereus isolate, jointly with the complete nhe operon and entfm gene (haplotype 5). As reported in Table 5, on considering the virulence profile alone, 22 (6.5%), 25 (7.3%) and 38 (11.2%) of the B. cereus isolates analyzed showed a profile of high, moderate, and low risk, respectively. ” Please refer to lines 249-266
POINT 11: The reference cited does not provide information on this topic. The definition seems quite arbitrary, why are strains harboring nhe genes and entFM (H4) more dangerous than strains with hbl and entFM (H6)? Strains able to produce complete NHE and/or HBL should be considered virulent, further differentiation needs testing for high and low toxin production.
Response 11: Thanks to thorough review, we stumbled upon a minor mistake in our manuscript. Therefore, some details are different in the categorization of the haplotypes, everything is highlighted in the current version. Please refer to lines 250-262.
POINT 12: Table 5: according to the above comment the “risk degrees” are not comprehensible, the data presented are not a risk assessment in the strict sense.
Response 12: As suggested by the reviewer the caption of the table 5 has been modified as:
“Number and percentage of water buffalo mozzarella cheese samples falling into the four risk categories (High, Moderate, Low and No Risk) according to the methods of determination adopted (pathogen contamination, virulence profile and a combined approach based on both bacterial contamination and virulence profile).
POINT 13: Conclusions are not supported by the results.
Response 13: The conclusion has been rewritten as:
“In conclusion, the consumer runs a moderate risk of buying water buffalo mozzarella cheese with a level of B. cereus high enough to cause food poisoning. However, the results of the present study showed that the risk of poisoning may increase during storage at room temperature, since the pathogen is able to grow rapidly. Moreover, a high percentage of the strains isolated harbored genes involved in toxin production, and a high percentage showed virulence properties associated with possible food-borne poisoning. However, on combining the findings of bacterial counting and analysis of the molecular profile, a different level of risk emerged. Specifically, a lower number of samples proved to be at high or moderate toxic risk. Furthermore, if the average per capita consumption of water buffalo mozzarella cheese is considered, the risk of poisoning by B. cereus is even lower.
Only a combined approach based on the B. cereus count and the molecular profiles of isolates can accurately assess the risk of poisoning. However, for a more complete evaluation, data on the typology of mozzarella consumers and the impact of B. cereus poisoning on human beings would be necessary. Indeed, the present study lacks quantitative information on the consequences of exposure. This is because the surveillance of enteric disorders in human beings is in some cases inadequate, especially with regard to self-limiting disorders, such as those caused by B. cereus.“ Please refer to lines 304-322

Reviewer 2 Report
The study is of major interest, but the English language/style is insufficient. This has to be significantly improved. The introduction must be optimized. Sometimes italic styles are missing. The introduction is missing accurate structuring.
The manuscript lacks carefulness.
The authors have missed directly detecting the described toxins. They only detected the toxin relevant gens. This has to be clearly addressed. Especially for cereulide, this is of major importance.
If the manuscript is significantly improved and the authors do rewrite the manuscript with more carefulness the reviewer would be disposed to re-evaluate the manuscript for publication.
The findings are interesting to the readers.
Author Response
Response to Reviewer 2 Comments
Point 1. The study is of major interest, but the English language/style is insufficient. This has to be significantly improved. The introduction must be optimized. Sometimes italic styles are missing. The introduction is missing accurate structuring.
The manuscript lacks carefulness.
The authors have missed directly detecting the described toxins. They only detected the toxin relevant gens. This has to be clearly addressed. Especially for cereulide, this is of major importance.
If the manuscript is significantly improved and the authors do rewrite the manuscript with more carefulness the reviewer would be disposed to re-evaluate the manuscript for publication.
The findings are interesting to the readers.
Response 1.
Dear Reviewer,
We thank for the valuable comments.
The manuscript has now been improved. We made the necessary adaptations and corrections throughout the manuscript (highlighted in yellow). Moreover, based on the reviewer suggestion we also have the paper revised by a native English speaker (highlighted in green).
Yours sincerely,
Yolande Therese Rose Proroga, corresponding author

Reviewer 3 Report
It is an interesting work on a very relevant food/hazard couple: B. cereus and Water Buffalo Mozarella cheese. It is one of the most consumed cheeses and yet few studies are devoted to this couple. The authors have chosen an original approach that provides both prevalence data of B. cereus and water buffalo mozarella cheese, and information on the toxinotype of strains isolated from commercial products. This approach allows them to determine 4 classes of risk (from high to no risk) and to assess consumer exposure in relation to the overall consumption of these products in Italy.
General comments : Cheese consumption data is very interesting but very limited. In particular, the typology of mozzarella consumers is not addressed or discussed, although this is a very important point to "appraise related risk" as you say. Do you have consumption data by sub-population (especially young people)? If not, can you discuss this in the revised version?
I think the format you used for the references is not correct (latin name, journal name italicized...).
There are a few points of detail to discuss or correct.
Line 2. I'm not sure for the word "assessment" in the title
Line 19 : 102 and 106 have to be corrected in 102 and 106
line 62 : delete 2. Material and Methods
line 58-62 : the sentence is very long and the end (i.e. ...to appraise the related risk.) seems to me a little exaggerated with regard to the work. Thank you for rewording.
line 72 : i think it's Saint Nom la Bretèche
line 166 : B. cereus in italic. Latin names are not always italicized in the text. Please check this carefully.
Author Response
Dear Reviewer,
We thank for the valuable comments.
We made the necessary adaptations and corrections throughout the manuscript (highlighted in yellow).
A detailed answer on each remark is included in the rebuttal. Moreover, based on the reviewer suggestion we also have the paper revised by a native English speaker (highlighted in green).
Yours sincerely,
Yolande Therese Rose Proroga, corresponding author
Response to Reviewer 3 Comments
Reviewer: It is an interesting work on a very relevant food/hazard couple: B. cereus and Water Buffalo Mozarella cheese. It is one of the most consumed cheeses and yet few studies are devoted to this couple. The authors have chosen an original approach that provides both prevalence data of B. cereus and water buffalo mozarella cheese, and information on the toxinotype of strains isolated from commercial products. This approach allows them to determine 4 classes of risk (from high to no risk) and to assess consumer exposure in relation to the overall consumption of these products in Italy.
POINT 1: General comments: Cheese consumption data is very interesting but very limited. In particular, the typology of mozzarella consumers is not addressed or discussed, although this is a very important point to "appraise related risk" as you say. Do you have consumption data by sub-population (especially young people)? If not, can you discuss this in the revised version?
Response 1: Thank you to point this out. Unfortunately, little information is available about the typology of mozzarella consumers. As suggested by the reviewer this it has now been discussed in the revised version. Please refer to lines 288-291.
POINT 2: I think the format you used for the references is not correct (latin name, journal name italicized...).
Response 2: Thank you for the suggestions. The references have been corrected.
POINT 3: Line 2. I'm not sure for the word "assessment" in the title
Response 3: "assessment" has been deleted in the title.
POINT 4: Line 19: 102 and 106 have to be corrected in 102 and 106
Response 4: “102 and 106” has been replaced with “102 and 106”. Please refer to line 53
POINT 5: line 62: delete 2. Material and Methods
Response 5: “2. Material and Methods” has been deleted. Please refer to line 76.
POINT 6: line 58-62: the sentence is very long and the end (i.e. ...to appraise the related risk.) seems to me a little exaggerated with regard to the work. Thank you for rewording.
Response 6: Thank you for this suggestion. The sentence has been modified as:
“Therefore, the aims of this study were: (i) to study the occurrence of B. cereus s.l. in WBMC, (ii) to evaluate the presence of nine genes encoding toxins in the isolated strains and (iii) to assess the risk of B. cereus poisoning due to the consumption of water buffalo mozzarella cheese through a combined approach based on the evaluation of bacterial contamination and analysis of the virulence profile.” Pease refer to lines 72-76
POINT 7: line 72: i think it's Saint Nom la Bretèche
Response 7: “Saint Nom la Bretèche” has been added in “(BagMixer®400 P, Interscience, Saint Nom la Bretèche, France)”. Please refer to line 87-88.
POINT 8: line 166: B. cereus in italic. Latin names are not always italicized in the text. Please check this carefully.
Response 8: this was indeed a mistake. we have now made these corrections along the whole manuscript

Reviewer 4 Report
Line 11: Bacillus cereus lacking italics. Correct to “Bacillus cereus”
Lines 12-13: no need for capital letter at the starting of each toxin. Correct to “associated with cytotoxin K, hemolysin BL, non-hemolytic enterotoxin and enterotoxin FM”.
Line 14: B. cereus lacking italics. Correct to “B. cereus”
Line 17: B. cereus lacking italics. Correct to “B. cereus”
Line 18: B. cereus lacking italics. Correct to “B. cereus”
Line 19: Missing superscript in CFU/g counts. Please update to 2.2x102 to 2.6x106 CFU/g.
Line 21: B. cereus lacking italics. Correct to “B. cereus”
Line 26: remove italics from “as”
Line 27-29: Missing B. cereus from the list of B. cereus sensu stricto (s.s). Please, add B. cereus to the list.
Line 42: Missing plural. Please update to read “ 105-108 cells or spores”
Line 62: Delete “2. Material and Methods”.
Line 117: Unnecessary Table S1. Bacterial growth per hour is reported as max. growth rate (h-1), max. rate (log.conc/h) or Dbl. time (hours). Both max. rate (log.conc/h) or Dbl. time (hours) are reported by ComBase. Max. growth rate (h-1) can be calculated from max. rate by multiplying it by ln(10).
Therefore, the authors should provide max. rate (log.conc/h) or Dbl. time (hours) for each of the study temperatures (15, 18, 22°C) together with the “Phys. State” (physical state of cells used for simulation) and the Lag time (hours) predicted by ComBase.
Lines 117-119: The authors should state the “Physic. State” used in simulations, as well as, elaborating the reason for selection of that specific value.
Line 120-122: Cell concentrations are reported with a single decimal. Lack of single decimal at time 0.
Line 166-167: B. cereus lacking italics. Correct to “B. cereus”
Line 173: This reviewer cannot see the relationship between the classification of isolates into hazard categories and reference (39). Could you please explain were in reference 39 is a similar classification present?
Line 174: delete “with” from sentence… “Whithin the with haplotypes…”
Line 190-193: Could the authors provide references for the relation between haplotypes and relationship to hazardousness? Because this reviewer could not see the relationship between the statement and the reference (36) provided.
Author Response
Dear Reviewer,
We thank for the valuable comments.
We made the necessary adaptations and corrections throughout the manuscript (highlighted in yellow).
A detailed answer on each remark is included in the rebuttal. Moreover, based on the reviewer suggestion we also have the paper revised by a native English speaker (highlighted in green).
Yours sincerely,
Yolande Therese Rose Proroga, corresponding author
Response to Reviewer 4 Comments
POINT 1: Line 11: Bacillus cereus lacking italics. Correct to “Bacillus cereus”
Response 1: this was indeed a mistake. we have now made these corrections along the whole manuscript
POINT 2: Lines 12-13: no need for capital letter at the starting of each toxin. Correct to “associated with cytotoxin K, hemolysin BL, non-hemolytic enterotoxin and enterotoxin FM”.
Response 2: The capital letters have been replaced with lowercase letters. Please refer to lines 13 and 14.
POINT 3: Line 14: B. cereus lacking italics. Correct to “B. cereus”
Response 3: We have now made these corrections along the whole manuscript
POINT 4: Line 17: B. cereus lacking italics. Correct to “B. cereus”
Response 4: We have now made these corrections along the whole manuscript
POINT 5: Line 18: B. cereus lacking italics. Correct to “B. cereus”
Response 5: We have now made these corrections along the whole manuscript
POINT 6: Line 19: Missing superscript in CFU/g counts. Please update to 2.2x102 to 2.6x106 CFU/g.
Response 6: “102 and 106” has been replaced with “102 and 106”. Please refer to line 53.
POINT 7: Line 21: B. cereus lacking italics. Correct to “B. cereus”
Response 7: We have now made these corrections along the whole manuscript.
POINT 8: Line 26: remove italics from “as”
Response 8: Italics has been removed from “as”. Please refer to line 30.
POINT 9: Line 27-29: Missing B. cereus from the list of B. cereus sensu stricto (s.s). Please, add B. cereus to the list.
Response 9: B. cereus has now been added to the list. Please refer to line 32.
POINT 10: Line 42: Missing plural. Please update to read “ 105-108 cells or spores”
Response 10: The correction has now been made. Please refer to line 53.
POINT 11: Line 62: Delete “2. Material and Methods”.
Response 11: “2. Material and Methods” has been deleted. Please refer to line 76.
POINT 12: Line 117: Unnecessary Table S1. Bacterial growth per hour is reported as max. growth rate (h-1), max. rate (log.conc/h) or Dbl. time (hours). Both max. rate (log.conc/h) or Dbl. time (hours) are reported by ComBase. Max. growth rate (h-1) can be calculated from max. rate by multiplying it by ln(10).
Therefore, the authors should provide max. rate (log.conc/h) or Dbl. time (hours) for each of the study temperatures (15, 18, 22°C) together with the “Phys. State” (physical state of cells used for simulation) and the Lag time (hours) predicted by ComBase.
Response 12: The information suggested by the reviewer have been now included both in Table S1 and in the text as: “Moreover, regarding microbial growth, the maximum growth rate (log.conc/h), the doubling time (hours), the maximum population density (MPD; log CFU/g) and the lag time (hours) were estimated (Table S1). On comparing the three temperatures, at 22°C B. cereus showed the highest growth rate per hour (0.268 log.conc/h), the shortest doubling time (1.125 hours) and the shortest lag time[time lag] (13.29 hours)..” Please refer to line 153-158.
POINT 13: Lines 117-119: The authors should state the “Physic. State” used in simulations, as well as, elaborating the reason for selection of that specific value.
Response 13: We have chosen the input parameter for the physiological state included in the ComBase Predictor (2.7 e-4). Thank you to point this out. The information has now been included in the text. Please refer to line 99-101.
POINT 14: Line 120-122: Cell concentrations are reported with a single decimal. Lack of single decimal at time 0.
Response 14: The decimal values have now been added. Please refer to lines 162-163.
POINT 15: Line 166-167: B. cereus lacking italics. Correct to “B. cereus”
Response 15: We have now made these corrections along the whole manuscript
POINT 16: Line 173: This reviewer cannot see the relationship between the classification of isolates into hazard categories and reference (39). Could you please explain were in reference 39 is a similar classification present?
Response 16: Thank you for the depth review. The reference indicated was indeed not correct, we have now included the right reference. Moreover, the sentences have been modified as “In the present study, the isolates were classified into hazard categories according to their molecular profile (Table 5). On the basis of the genes harbored, six haplotypes (H1-H6) showed virulence properties related to possible food-borne poisoning, as reported by [36].” Please refer to lines 249-252.
POINT 17: Line 174: delete “with” from sentence… “Whithin the with haplotypes…”
Response 17: The sentences has been modified as “On the basis of the genes harbored, six haplotypes (H1-H6) showed virulence properties related to possible food-borne poisoning, as reported by [36]” Please refer to lines 250-252.
POINT 18: Line 190-193: Could the authors provide references for the relation between haplotypes and relationship to hazardousness? Because this reviewer could not see the relationship between the statement and the reference (36) provided.
Response 18: Thank you for the depth review. The reference indicated was indeed not correct and we have now included the right reference. Moreover, the text has been modified as: “In this analysis, the “high risk” category comprised samples with B. cereus contamination above 103 CFU/g - an infective dose considered unsafe for human consumption [37] - and high-risk toxinotype strains (haplotype 1), and those with higher B. cereus contamination (>105 CFU/g) but lower molecular risk profiles of the isolates (haplotypes 2 and 3). The “moderate risk” group comprised samples displaying a higher risk of toxic strains (haplotype 1) but lower bacterial contamination, and samples with an intermediate level of contamination (from 3 to 5 Log CFU/g) and haplotypes 2 and 3. All the remaining samples were classified as “no risk”, as shown in Table 5.” Please refer to lines 270-279.

Round 2
Reviewer 1 Report
The manuscript has been partly improved.
There is, however, still a major issue with the definition of the “hazard categories”. The definition is still quite arbitrary.
The authors state: “On the basis of the genes harbored, six haplotypes (H1-H6) showed virulence properties related to possible food-borne poisoning, as reported by [36]." I fully agree with this sentence!
But then, what is the scientific evidence for the different levels of risk attributed to H1, H2/3 and H4-6, respectively, in Table 5 and discussion? There is still no reference given supporting this hypothesis. Strains able to produce complete NHE and/or HBL have been involved in food poisoning and should be considered virulent, further differentiation needs testing for high and low toxin production.
I strongly recommend to either provide reference for this claim or to modify/skip this part.
There are still some minor inconsistencies in denomination of genes.
Author Response
Dear Reviewer,
We thank for the valuable comments.
We made the necessary adaptations and corrections throughout the manuscript (highlighted in yellow).
Yours sincerely,
Yolande Therese Rose Proroga, corresponding author
Response to Reviewer 1 Comments
The manuscript has been partly improved.
There is, however, still a major issue with the definition of the “hazard categories”. The definition is still quite arbitrary.
POINT 1: The authors state: “On the basis of the genes harbored, six haplotypes (H1-H6) showed virulence properties related to possible food-borne poisoning, as reported by [36]." I fully agree with this sentence!
But then, what is the scientific evidence for the different levels of risk attributed to H1, H2/3 and H4-6, respectively, in Table 5 and discussion? There is still no reference given supporting this hypothesis. Strains able to produce complete NHE and/or HBL have been involved in food poisoning and should be considered virulent, further differentiation needs testing for high and low toxin production.
I strongly recommend to either provide reference for this claim or to modify/skip this part.
Response 1: We performed the categorization based on the presence or absence of the different genes recognized as playing a major role in the disease. Basing of the reviewer recommendation we have now included more references and we have modified the manuscript as:
“Based on the presence or absence of the different genes recognized as playing a major role in the disease, we have categorized the haplotypes in three groups on the basis of the risk for B. cereus disease: haplotypes at "high risk" , “moderate risk” and “low risk”. The haplotype H1 harboring nhe (A, B, C), hbl (A, C, D), cytK and entFM genes has been considered the haplotype having the greatest risk (“High Risk”) to cause enteric disorders.
Isolates characterized by the haplotypes H2 [nhe (A,B,C); hbl (A,C,D)] and H3 [nhe (A,B,C); cytK; entFM] showed a “Moderate Risk” owing to the presence of the nhe and hbl genes in H2 and the presence of nhe and cytK genes in H3. However, compared with H1, the risk of enteric disorders arising from the consumption of foods contaminated with H2 and H3 is lower for the absence of cytK gene which toxin is highly toxic towards human intestinal epithelial cells [37], or hbl genes. Indeed, HBL complex is one of the most important responsible for enterotoxigenic activity of B. cereus strains, thus its absence lead to a reduction of the cytotoxic and hemolytic activity of this pathogen [38,39]. Haplotypes H4 [nhe (A,B,C); entFM], H5 [nhe (A,B,C); entFM; ces] and H6 [nhe (A,C); hbl (A,C,D); entFM] showed a lower level of risk (“Low Risk”), owing to the absence in all of them of the cytK gene and the absence either of genes coding for HBL toxin components or of the nheB gene. H6, which was infrequently detected (5.6%), harbored the incomplete nhe operon; it was therefore regarded as posing a low risk, since all three components (A, B and C) of the nhe operon are needed for full toxicity [11]. The ces gene was detected in only one B. cereus isolate, jointly with the complete nhe operon and entFM gene (haplotype 5). As reported in Table 5, on considering the virulence profile alone, 22 (6.5%), 25 (7.3%) and 38 (11.2%) of the B. cereus isolates analyzed showed a profile of high, moderate, and low risk, respectively.”
Please refer to lines: 254-278
POINT 2: There are still some minor inconsistencies in denomination of genes.
Response 2: We have now made these corrections along the whole manuscript

Reviewer 2 Report
A direct response to the reviewer's points would have been better. Nevertheless, the manuscript is improved, but still, you don't address the missing toxin detection methods. It is possible that the bacteria are gone bu,t the toxin (cereulide) is still present. Thus, such a conclusion: "Specifically, a lower number of samples proved to be at high or moderate toxic risk" cant be judged if you don't look for the toxins.
So please once, again:
The authors have missed directly detecting the described toxins. They only detected the toxin relevant gens. This has to be clearly addressed. Especially for cereulide, this is of major importance.
Please address this; if not, you will mislead the reader who is not so much into the topic.
Author Response
Dear Reviewer,
We thank for the valuable comments.
We made the necessary adaptations and corrections throughout the manuscript (highlighted in yellow).
Yours sincerely,
Yolande Therese Rose Proroga, corresponding author
Response to Reviewer 2 Comments
POINT 1: A direct response to the reviewer's points would have been better. Nevertheless, the manuscript is improved, but still, you don't address the missing toxin detection methods. It is possible that the bacteria are gone bu,t the toxin (cereulide) is still present. Thus, such a conclusion: "Specifically, a lower number of samples proved to be at high or moderate toxic risk" cant be judged if you don't look for the toxins.
So please once, again:
The authors have missed directly detecting the described toxins. They only detected the toxin relevant gens. This has to be clearly addressed. Especially for cereulide, this is of major importance.
Please address this; if not, you will mislead the reader who is not so much into the topic.
Response 1: We are completely agree with the reviewer. We have performed only the detection of the genes but for a complete virulence assessment further studies should focus also on the detection of the toxins (especially for cereulide). Basing on the reviewer suggestion we have now stressed this concept in the manuscript. Please refer to lines 295-299 and 342-344.

Round 3
Reviewer 2 Report
Thank you very much for the improvments.